# A Simple and Optimal Policy Design for Online Learning with Safety against Heavy-tailed Risk

**David Simchi-Levi**
Institute for Data, Systems, and Society
Massachusetts Institute of Technology
Cambridge, MA 02139
dslevi@mit.edu

**Zeyu Zheng**
Industrial Engineering & Operations Research
University of California, Berkeley
Berkeley, CA 94720
zyzheng@berkeley.edu

**Feng Zhu**
Institute for Data, Systems, and Society
Massachusetts Institute of Technology
Cambridge, MA 02139
fengzhu@mit.edu

## Abstract

We consider the classical multi-armed bandit problem and design simple-to-implement new policies that simultaneously enjoy two properties: worst-case optimality for the expected regret, and safety against heavy-tailed risk for the regret distribution. Recently, [10] showed that information-theoretic optimized bandit policies as well as standard UCB policies suffer from some serious heavy-tailed risk; that is, the probability of incurring a linear regret slowly decays at a polynomial rate of $1/T$, as $T$ (the time horizon) increases. Inspired by their result, we further show that any policy that incurs an instance-dependent $O(\ln T)$ regret must incur a linear regret with probability $\Omega(\mathrm{poly}(1/T))$ and that the heavy-tailed risk actually exists for all "instance-dependent consistent" policies. Next, for the two-armed bandit setting, we provide a simple policy design that (i) has the worst-case optimality for the expected regret at order $\tilde{O}(\sqrt{T})$ and (ii) has the worst-case tail probability of incurring a linear regret decay at an exponential rate $\exp(-\Omega(\sqrt{T}))$. We further prove that this exponential decaying rate of the tail probability is optimal across all policies that have worst-case optimality for the expected regret. Finally, we generalize the policy design and analysis to the general setting with an arbitrary $K$ number of arms. We provide detailed characterization of the tail probability bound for any regret threshold under our policy design. Numerical experiments are conducted to illustrate the theoretical findings. Our results reveal insights on the incompatibility between consistency and light-tailed risk, whereas indicate that worst-case optimality on expected regret and light-tailed risk are compatible.

## 1 Introduction

The stochastic multi-armed bandit (MAB) problem is a widely studied problem in the domain of sequential decision-making under uncertainty, with many applications such as online advertising, recommendation systems, clinical trials, financial portfolio design, etc. It also has valuable theoretical insights exhibiting the exploration-exploitation trade-off For policy design and analysis, much of the MAB literature uses the metric of maximizing the *expected* cumulative reward, or equivalently minimizing the *expected regret* (where *regret* is defined as the difference between the cumulative reward obtained by always pulling the best arm and by executing a policy that does not a priori

36th Conference on Neural Information Processing Systems (NeurIPS 2022).

know the reward distributions). The optimality of a policy is often characterized through its expected regret's rate (order of dependence) on the experiment horizon $T$.

In this paper, however, we show that renowned policies (such as the standard Upper Confidence Bound (UCB) policy, the Successive Elimination (SE) policy and the Thompson Sampling (TS) policy) that are designed to enjoy optimality in terms of expected regret can incur a "heavy-tailed risk". That is, the distribution of the regret has a heavy tail — the probability of incurring a linear regret slowly decays at a polynomial rate $\Omega(\text{poly}(1/T))$ as $T$ tends to infinity. In contrast, a "light-tailed" risk means that the probability of a policy incurring a linear regret decays at an exponential rate $\exp(-\Omega(T^\beta))$ for some $\beta > 0$. The heavy-tailed risk can be undesired when an MAB policy is used in applications that are sensitive to tail risks (e.g., finance, healthcare, supply chain). Our work is primarily motivated by trying to answer the following question. ***Can we design a simple policy that on one hand enjoys the optimality under the expected regret notion, whereas on the other hand has light-tailed risk?*** The results of our work are summarized as follows.

**1. Instant-dependent consistency and light-tailed risk are incompatible.** Inspired by the results in [8], we adapt their change of measure argument to show that any policy that incurs an instance-dependent $O(\ln T)$ regret must incur a linear regret with probability $\Omega(\text{poly}(1/T))$. We additionally show that any instant-dependent consistent policy cannot be light-tailed: if an instant-dependent consistent policy has the probability of incurring a linear regret decay as $\exp(-f(T))$, then $f(T)$ must be $o(T^\beta)$ as $T \to +\infty$ for any $\beta > 0$.

**2. Worst-case optimality and light-tailed risk can co-exist.** Starting from the two-armed bandit setting, we provide a simple policy design and prove that it enjoys both the worst-case optimality $\tilde{O}(\sqrt{T})$ for the expected regret and the light-tailed risk $\exp(-\Omega(\sqrt{T}))$ for the regret distribution. We also show that such exponential decaying rate of the tail probability is optimal within the class of worst-case optimal policies.

**3. Extensions and Experiments.** We extend the result from the two-armed bandit to the general $K$-armed bandit and characterize the tail probability bound for any regret threshold in an explicit form. We prove that under our policy design, the worst-case probability of incurring a regret larger than $x$ is bounded by $\exp(-\Omega(x/K\sqrt{T}))$. We find that the associated proof techniques are new and may benefit broader analysis on regret distribution and tail risk. Our result also partially solves an open problem raised in [14]. Experiments are also conducted to illustrate our theoretical findings.

## 1.1 Related Work

Our work builds upon the vast literature of designing and analyzing policies for the multi-armed bandit (MAB) problem and its various extensions. A comprehensive review can be found in [6, 17, 14]. A standard paradigm for obtaining a near-optimal regret is to first *fix* some confidence parameter $\delta > 0$. Then a "good event" is defined such that good properties are retained conditioned on the event (for example, in the MAB problem, the good event is such that the mean of each arm always lies in the confidence bound). Then one can obtain both high-probability and worst-case expected regret bounds through careful analysis on the good event. It is known that the MAB problem has the following regret bound: for any fixed $\delta \in (0, 1)$, the regret bound of UCB is bounded by $O(\sqrt{KT \ln(T/\delta)})$ with probability at least $1 - \delta$. Or equivalently speaking, the probability of incurring a $\Omega(\sqrt{KT \ln(T/\delta)})$ regret is bounded by $\delta$. However, the parameter $\delta$ must be an input parameter for the policy. We will discuss this issue in more details in Section 3. In Section 17.1 of [14], an open question is asked: is it possible to design a single policy such that the worst-case probability of incurring a $\Omega(\sqrt{KT} \ln(1/\delta))$ regret is bounded by $\delta$ for *any* $\delta > 0$? We partially solve this question by designing a policy such that for *any* $\delta > 0$, the probability of incurring a $\Omega(K\sqrt{T}(\sqrt{\ln T} + \ln(1/\delta)))$ regret is bounded by $\delta$.

There has been not much work on understanding the tail risk of bandit algorithms. Our work is inspired by [10]. They showed that optimized UCB bandit designs are fragile to mis-specifications and they modified UCB algorithms to ensure a desired polynomial rate of tail risk, which makes the algorithms more robust to mis-specifications. Different from their work, we propose a simple and new policy design that leads to both light-tailed risk (tail bound exponentially decaying with $\sqrt{T}$) and worst-case optimality (expected regret bounded by $\tilde{O}(\sqrt{T})$). Besides [10], two earlier works are [5, 15] and they study the concentration properties of the regret around the instance-dependent mean $O(\ln T)$. They show that in general the regret of the policies concentrate only at a polynomial

rate. That is, the probability of incurring a regret of $c(\ln T)^p$ (with $c > 0$ and $p > 1$ fixed) is approximately polynomially decaying with $T$. Different from our work, the concentration in their work is under an instance-dependent environment, and so such polynomial rate might be different across different instances. Nevertheless, their results indicate that standard bandit algorithms generally have undesirable concentration properties. Recently, [4] show that an online learning policy with the goal of obtaining logarithmic regret can be fragile, in the sense that a mis-specified risk parameter (e.g., the parameter for subgaussian noises) in the policy can incur an instance-dependent expected regret polynomially dependent on $T$. They then focus on robust algorithm development to circumvent the issue. Note that their goal is to handle mis-specification related with risk, but still the task is to minimize the expected regret.

Very recently, there are some works trying to understand the behaviour of UCB and TS policies by considering the diffusion approximations (see, e.g., [3, 19, 9, 12]). Although some distributional characterizations of regret are obtained in these works, the asymptotic regime is typically set such that the gaps between arm means shrink with the total time horizon. This makes their results incomparable to ours because we study how the tail probability decays with $T$ under unchanged environments. Another line of works closely related with ours involve solving risk-averse formulations of the MAB problem (see, e.g., [16, 11, 20, 18, 8, 13]). Compared to standard MAB problems, the main difference in their works is that arm optimality is defined using formulations other than the expected value, such as mean-variance criteria and (conditional) value-at-risk. These formulations consider some single metric that is different compared to the expected regret. From the formulation perspective, our work is different in the sense that we develop policies that on one hand maintain the classical worst-case optimal expected regret, whereas simultaneously achieve light-tailed risk bound. The policy design and analysis in our work are therefore also different from the literature.

## 2 Setup and Notation

The setting is described as follows. Fix a time horizon of $T$ and the number of arms as $K$. In each time $t \in [T]$, the decision maker (DM) needs to decide which arm $A_t \in [K]$ should be pulled. To be more precise, let $H_t = \{A_1, X_1, \cdots, A_{t-1}, X_{t-1}\}$ be the history prior to time $t$. When $t = 1$, $H_1 = \emptyset$. In time $t$, the DM adopt a policy $\pi_t : H_t \longmapsto A_t$ that maps the history $H_t$ to an action $A_t$, where $A_t$ follows a discrete probability distribution on $[K]$ determined by $H_t$. The environment then independently samples a reward $r_{t,A_t} = \theta_{A_t} + \epsilon_{t,A_t}$ and reveals it to the DM. Here, $\theta_{A_t}$ is the mean reward of arm $A_t$, and $\epsilon_{t,A_t}$ is an independent zero-mean noise term. We assume that $\epsilon_{t,A_t}$ is $\sigma$-subgaussian. That is, there exists a $\sigma > 0$ such that for any time $t$ and arm $k$,

$$\max\{\mathbb{P}(\epsilon_{t,k} \geq x), \mathbb{P}(\epsilon_{t,k} \leq -x)\} \leq \exp(-x^2/(2\sigma^2)).$$

Let $\theta = (\theta_1, \cdots, \theta_K)$ be the mean vector. Let $\theta_* = \max\{\theta_1, \cdots, \theta_K\}$ be the optimal mean reward among the $K$ arms. Note that DM does not know $\theta$ at the beginning, except that $\theta \in [0, 1]^K$. The empirical regret of the policy $\pi = (\pi_1, \cdots, \pi_T)$ under the mean vector $\theta$ over a time horizon of $T$ is defined as

$$\hat{R}_\theta^\pi(T) = \theta_* \cdot T - \sum_{t=1}^{T}(\theta_{A_t} + \epsilon_{t,A_t}).$$

Let $\Delta_k = \theta_* - \theta_k$ be the gap between the optimal arm and the $k$th arm. Let $n_{t,k}$ be the number of times arm $k$ has been pulled up to time $t$. That is, $n_{t,k} = \sum_{s=1}^{t} \mathbb{1}\{A_s = k\}$. For simplicity, we will also use $n_k = n_{T,k}$ to denote the total number of times arm $k$ is pulled throughout the whole time horizon. We define $t_k(n)$ as the time period that arm $k$ is pulled for the $n$th time. Define the pseudo regret as

$$R_\theta^\pi(T) = \sum_{k=1}^{K} n_k \Delta_k$$

and the genuine noise as

$$N^\pi(T) = \sum_{t=1}^{T} \epsilon_{t,A_t} = \sum_{k=1}^{K} \sum_{m=1}^{n_k} \epsilon_{t_k(m),k}.$$

Then the empirical regret can also be written as $\hat{R}_\theta^\pi(T) = R_\theta^\pi(T) - N^\pi(T)$. The following simple lemma gives the mean and the tail probability of the genuine noise $N^\pi(T)$. Intuitively, it shows

when bounding the mean or the tail probability of the empirical regret, one only need to consider the pseudo regret term. We will make it more precise when we discuss the proof of main theorems.

**Lemma 1** *We have* $\mathbb{E}[N^\pi(T)] = 0$ *and*

$$\max\left\{\mathbb{P}\left(N^\pi(T) \geq x\right), \mathbb{P}\left(N^\pi(T) \leq -x\right)\right\} \leq \exp\left(\frac{-x^2}{2\sigma^2 T}\right).$$

Before proceeding, we introduce some other notations. Throughout the paper, we use $O(\cdot)$ $(\tilde{O}(\cdot))$ and $\Omega(\cdot)$ $(\tilde{\Omega}(\cdot))$ to present upper and lower bounds on the growth rate up to constant (logarithmic) factors, and $\Theta(\cdot)$ $(\tilde{\Theta}(\cdot))$ to characterize the rate when the upper and lower bounds match up to constant (logarithmic) factors. We use $o(\cdot)$ to present strictly dominating upper bounds. In addition, for any $a, b \in \mathbb{R}$, $a \wedge b = \min\{a, b\}$ and $a \vee b = \max\{a, b\}$. For any $a \in \mathbb{R}$, $a_+ = \max\{a, 0\}$.

## 2.1 Light-tailed Risk, Instance-dependent Consistency, Worst-case Optimality

Now we discuss several core concepts that we are interested in.

**1. Light-tailed risk.** A policy is called light-tailed, if for any constant $c > 0$, there exists some $\beta > 0$ and constant $C > 0$ such that

$$\limsup_{T \to +\infty} \frac{\ln\left\{\sup_\theta \mathbb{P}\left(\hat{R}_\theta^\pi(T) > cT\right)\right\}}{T^\beta} \leq -C.$$

Note that here, we allow $\beta$ and $C$ to be dependent on $c$. In brief, a policy has light-tailed risk if the probability of incurring a linear regret can be bounded by an exponential term of polynomial $T$:

$$\sup_\theta \mathbb{P}(\hat{R}_\theta^\pi(T) \geq cT) = \exp(-\Omega(T^\beta))$$

for some $\beta > 0$. If a policy is not light-tailed, we say it is *heavy-tailed*.

**2. Instance-dependent consistency.** A policy is called consistent, if for any underlying true mean vector $\theta$, the policy has that

$$\limsup_{T \to +\infty} \frac{\mathbb{E}\left[\hat{R}_\theta^\pi(T)\right]}{T^\beta} = 0$$

holds for any $\beta > 0$. We emphasize that here "instant-dependent" means the formula above holds for any *fixed* environment parameter $\theta$. In brief, a policy is consistent if the expected regret can never be polynomially growing in $T$ for any fixed instance.

**3. Worst-case optimality.** A policy is said to be worst-case optimal, if for any $\beta > 0$, we have

$$\limsup_{T \to +\infty} \frac{\sup_\theta \mathbb{E}\left[\hat{R}_\theta^\pi(T)\right]}{T^{1/2+\beta}} = 0.$$

In brief, a policy is worst-case optimal if the worst-case expected regret can never be growing in a polynomial rate faster than $T^{1/2}$. Note that here we adopt a relaxed definition of optimality, in the sense that we do not clarify how the regret scale with the number of arms $K$ compared to that in literature. The notion of worst-case optimality in this work focuses on the dependence on $T$. For example, a policy with worst-case regret $O(\text{poly}(K)\sqrt{T} \cdot \text{poly}(\ln T))$ is also optimal by our definition.

It is well known that for the multi-armed bandit problem, one can design algorithms to achieve both instance-dependent consistency and worst-case optimality. Among them, two types of policies are of prominent interest: Successive Elimination (SE) and Upper Confidence Bound (UCB). We list the algorithm paradigms in Algorithm 1 and 2. The bonus term $\text{rad}(n)$ is typically set as

$$\text{rad}(n) = \sigma\sqrt{\frac{\eta \ln T}{n}} \tag{1}$$

with $\eta > 0$ being some tuning parameter.

| **Algorithm 1** Successive Elimination |
| --- |
| 1: $\mathcal{A} = [K]$. $t \leftarrow 0$. |
| 2: **while** $t < T$ **do** |
| 3:     Pull each arm in $\mathcal{A}$ once. $t \leftarrow t + K$. |
| 4:     Eliminate any $k \in \mathcal{A}$ from $\mathcal{A}$ if |
|     $\exists k' : \hat{\mu}_{t,k'} - \mathrm{rad}(n_{t,k'}) > \hat{\mu}_{t,k} + \mathrm{rad}(n_{t,k})$ |
| 5: **end while** |

| **Algorithm 2** Upper Confidence Bound |
| --- |
| 1: $\mathcal{A} = [K]$. $t \leftarrow 0$. |
| 2: **while** $t < T$ **do** |
| 3:     $t \leftarrow t + 1$. |
| 4:     Pull the arm with the highest UCB: |
|     $\mathrm{UCB}_{t-1,k} = \hat{\mu}_{t-1,k} + \mathrm{rad}(n_{t-1,k})$. |
| 5: **end while** |

## 3 Main Results: Two-armed Bandit

We start from the simple two-armed bandit setting. The general multi-armed setting is deferred to the next section. We first show that standard policies considered widely in the literature are heavy-tailed. The result reveals a fundamental incompatibility between consistency and light-tailed risk. Then we show how to add a simple twist to standard confidence bound based policies to obtain light-tailed policies. Moreover, we show that our design leads to an optimal tail decaying rate.

### 3.1 Instance-dependent Consistency Causes Heavy-tailed Risk

**Theorem 1** *If a policy $\pi$ is instance-dependent consistent, then it can never be light-tailed. Moreover, if $\pi$ satisfies*

$$\limsup_{T \to +\infty} \frac{\mathbb{E}\left[\hat{R}_\theta^\pi(T)\right]}{\ln T} < +\infty, \tag{2}$$

*then we have the following stronger argument. For any $c > 0$, there exists $C_\pi > 0$ such that*

$$\liminf_{T \to +\infty} \frac{\ln\left\{\sup_\theta \mathbb{P}\left(\hat{R}_\theta^\pi(T) > cT\right)\right\}}{\ln T} \geq -C_\pi. \tag{3}$$

Theorem 1 suggests that a consistent policy must have a risk tail heavier than an exponential one. The proof of Theorem 1 relies on a change of measure argument appeared in [10]. We consider the environment where the noise $\epsilon$ is Gaussian with standard deviation $\sigma$. Intuitively speaking, if we want a policy to be adaptive enough to handle different instances, then the policy will be sensitive to risk. Moreover, if the policy achieves $O(\ln T)$ regret for any fixed instance $\theta$ (the constant is typically dependent on $\theta$), then the probability of incurring a linear regret becomes $\exp(-O(\ln T)) = \Omega(\mathrm{poly}(1/T))$. One special case is the family of confidence bound related policies (SE and UCB) where from Theorem 1 one can see that the standard bonus term (1) will always lead to a tail polynomially dependent on $T$. Another example is the Thompson Sampling (TS) policy. It has been established that $\pi = \mathrm{TS}$ with Beta or Gaussian priors has the property (2) (see, e.g., Theorem 1 and 2 in [1], proof of Theorem 1.3 in [2]). Theorem 1 then suggests that (3) also holds for $\pi = \mathrm{TS}$.

We need to remark on the difference between Theorem 1 and high-probability bounds in previous literature. It has been well-established that UCB or SE with

$$\mathrm{rad}(n) = \sigma\sqrt{\frac{\eta \ln(1/\delta)}{n}}$$

achieves $\tilde{O}(\sqrt{T \cdot \mathrm{polylog}(T/\delta)})$ regret with probability at least $1 - \delta$ (see, e.g., Section 1.3 in [17], Section 7.1 in [14]). Such design also leads to a consistent policy. However, the bound holds only for fixed $\delta$. In fact, one can see that the bonus design is dependent on the confidence parameter $\delta$. If $\delta = \exp(-\Omega(T^\beta))$ with $\beta > 0$, then the scaling speed of the expected regret with respect to $T$ can only be greater than $1/2$, which is sub-optimal. As a comparison, in our problem, ideally we seek to find a single policy such that it achieves $\tilde{O}(\sqrt{T \cdot \mathrm{polylog}(T/\delta)})$ regret for *any $\delta > 0$*.

Up till now, we make two core observations. First, from standard MAB results, consistency and optimality can hold simultaneously. Second, from Theorem 1, consistency and light-tailed risk

are always incompatible. Then a natural question arises: Can we design a policy that enjoys both optimality and light-tailed risk? If we can, then can we make the tail risk decay with $T$ in an optimal rate? We answer these two questions with an affirmative "yes" in the next section.

## 3.2   A New Policy Design Achieving Light-tailed Risk and Worst-case Optimality

In this section, we propose a new policy design that achieves both light-tailed risk and worst-case optimality. The design is simple. We still use the idea of confidence bounds, but instead of setting the bonus as (1), we set

$$\text{rad}(n) = \sigma \frac{\sqrt{\eta T \ln T}}{n} \tag{4}$$

with $\eta > 0$ being a tuning parameter. Theorem 2 gives performance guarantees for the mean and the tail probability of the empirical regret when $\pi = \text{SE}$.

**Theorem 2** *For the two-armed bandit problem, the SE policy with $\eta \geq 4$ and the bonus term being* (4) *satisfies the following two properties. 1.* $\sup_\theta \mathbb{E}[\hat{R}_\theta^\pi] = O(\sqrt{T \ln T})$. *2. For any $c > 0$ and any* $\alpha \in (1/2, 1]$, *we have*

$$\sup_\theta \mathbb{P}(\hat{R}_\theta^\pi(T) \geq cT^\alpha) = \exp(-\Omega(T^{\alpha - 1/2})).$$

The first item in Theorem 2 means that with the modified bonus term, the worst case regret is still bounded by $O(\sqrt{T \ln T})$, which is the same as the regret bounds for SE and UCB with the standard bonus term (1). The second item shows that the tail probability of incurring a $\Omega(T^\alpha)$ regret ($\alpha > 1/2$) is exponentially decaying in $\Omega(T^{\alpha - 1/2})$, and thus the policy is light-tailed. The detailed proof of Theorem 2 is provided in the supplementary material. An illustrative road-map of the proof is delegated to Section 4, where we provide the proof idea for a theorem that is a strict generalization of Theorem 2. Here, we give some intuition on the new bonus design. Our new bonus term inflates the standard one by a factor of $\sqrt{T/n}$. This means our policy is more conservative than the traditional confidence bound methods, especially at the beginning. In fact, one can observe that for the first $\Theta(\sqrt{T})$ time periods, our policy consistently explores between arm 1 and 2, regardless of the environment. A natural corollary is that our policy is never "consistent", and this is reasonable following Theorem 1. However, the bonus term (4) decays at a faster rate on the number of pulling times $n$ compared to (1). This means as the experiment goes on, the policy leans towards exploitation. We need to stress that this is not the same as the explore-then commit policy, which is well-known to achieve a sub-optimal $\Theta(T^{2/3})$ regret (see, e.g., Section 1.2 in [17], Section 6.1 in [14]).

The following theorem shows that the risk tail in Theorem 2 is not improvable. That is, if the policy $\pi$ is worst-case optimal, then for fixed $\alpha \in (1/2, 1]$, the exponent of $\alpha - 1/2$ is tight.

**Theorem 3** *Let $c \in (0, 1/2)$. Consider the 2-armed bandit problem with Gaussian noise. Let $\pi$ be a worst-case optimal policy. That is, for any $\alpha > 1/2$,*

$$\limsup_T \frac{\sup_\theta \mathbb{E}[\hat{R}_\theta^\pi(T)]}{T^\alpha} = 0.$$

*Then for any $\alpha \in (1/2, 1]$,*

$$\liminf_T \frac{\ln\left\{\sup_\theta \mathbb{P}(\hat{R}_\theta^\pi(T) \geq cT^\alpha)\right\}}{T^\beta} = 0$$

*holds for any $\beta > \alpha - 1/2$.*

Theorem 3 also relies on the change of measure argument appeared in Theorem 1. However, there are two important differences: we only have worst-case optimality rather than consistency, and the regret threshold $cT^\alpha$ is in general not linear in $T$. Therefore, we need to take care of constructing the specific "hard" instance when doing the change of measure. The detailed proof is delegated to the supplementary material.

# 4 Extensions: Multi-armed Bandit

In this section, we provide an extension of our previous tail probability bound in Theorem 2 from the following three aspects: (a) a general $K$-armed bandit setting; (b) an analysis for UCB aside from SE; (c) a detailed characterization of the tail bound for any fixed regret threshold.

**Theorem 4** *For the $K$-armed bandit problem, $\pi = \text{SE}$ or $\pi = \text{UCB}$ with*

$$\text{rad}(n) = \sigma \frac{\sqrt{\eta T \ln T}}{n}$$

*satisfy the following two properties. 1. If $\eta \geq 4$, then $\sup_\theta \mathbb{E}\left[R_\theta^\pi\right] \leq 4K + 4K\sigma\sqrt{\eta T \ln T}$. 2. If $\eta > 0$, then for any $x > 0$, we have*

$$\sup_\theta \mathbb{P}(R_\theta^\pi(T) \geq x)$$

$$\leq \exp\left(-\frac{x^2}{2K\sigma^2 T}\right) + 2K \exp\left(-\frac{(x - 2K - 4K\sigma\sqrt{\eta T \ln T})_+^2}{32\sigma^2 K^2 T}\right) + K^2 T \exp\left(-\frac{x\sqrt{\eta \ln T}}{8\sigma K\sqrt{T}}\right).$$

**Proof Idea.** We provide a road-map of proving Theorem 4. The expected regret bound is proved using standard techniques. That is, we define the good event to be such that the mean of each arm always lies in the confidence bounds throughout the whole time horizon. Conditioned on the good event, the regret of each arm is bounded by $O(\sqrt{T \ln T})$, and thus the total expected regret is $O(K\sqrt{T \ln T})$.

The proof of the tail bound requires more effort. Without loss of generality, we assume arm 1 is optimal. We first illustrate the proof for $\pi = \text{SE}$.

*Step 1.* We use

$$\sup_\theta \mathbb{P}(\hat{R}_\theta^\pi(T) \geq x) \leq \mathbb{P}\left(N^\pi(T) \leq -x/\sqrt{K}\right) + \sup_\theta \mathbb{P}\left(R_\theta^\pi(T) \geq x(1 - 1/\sqrt{K})\right)$$

The term with the genuine noise can be easily bounded using Lemma 1. We are left to bound the tail of the pseudo regret. By a union bound, we observe that

$$\mathbb{P}\left(R_\theta^\pi(T) \geq x(1 - 1/\sqrt{K})\right) \leq \sum_{k \neq 1} \mathbb{P}\left(n_k \Delta_k \geq x/(K + \sqrt{K})\right) \leq \sum_{k \neq 1} \mathbb{P}\left(n_k \Delta_k \geq x/(2K)\right)$$

Thus, we reduce bounding the sum of the regret incurred by different arms to bounding that by a single sub-optimal arm.

*Step 2.* For any $k \neq 1$, we define

$$S_k = \{\text{Arm 1 is not eliminated before arm } k\}.$$

With a slight abuse of notation, we let $n_0 = \lceil x/(2K\Delta_k) \rceil - 1$. If both $n_k \Delta_k \geq x/(2K)$ and $S_k$ happen, then arm 1 and $k$ are not eliminated after each of them being pulled $n_0$ times. This indicates

$$\hat{\mu}_{t_1(n_0),1} - \frac{\sigma\sqrt{\eta T \ln T}}{n_0} \leq \hat{\mu}_{t_k(n_0),k} + \frac{\sigma\sqrt{\eta T \ln T}}{n_0}$$

The probability of this event can be bounded using concentration of subgaussian variables, which yields the second term in the tail bound in Theorem 4. The choice of $n_0$ is important. We note that at this step, even if we replace our new bonus term by the standard one, the bound still holds.

*Step 3.* Now consider the situation when $\bar{S}_k$ happens. This means after some phase $n$, the optimal arm 1 is eliminated by some arm $k'$, while arm $k$ is not eliminated. Note that $k = k'$ does not necessarily hold when $K > 2$. As a consequence, we have the following two events hold simultaneously:

$$\hat{\mu}_{t_{k'}(n),k'} - \frac{\sigma\sqrt{\eta T \ln T}}{n} \geq \hat{\mu}_{t_1(n),1} + \frac{\sigma\sqrt{\eta T \ln T}}{n}, \hat{\mu}_{t_k(n),k} + \frac{\sigma\sqrt{\eta T \ln T}}{n} \geq \hat{\mu}_{t_1(n),1} + \frac{\sigma\sqrt{\eta T \ln T}}{n}.$$

The first event leads to

$$\text{Mean of some noise terms} \geq \frac{2\sigma\sqrt{\eta T \ln T}}{n} + \Delta_{k'} \geq \frac{2\sigma\sqrt{\eta T \ln T}}{n}.$$

The second event leads to

$$\text{Mean of some noise terms } \geq \Delta_k \geq \frac{x}{2KT}.$$

Now comes the trick to deal with an arbitrary $n$. We bound the probabilities of the two events separately and take the minimum of the two probabilities ($\mathbb{P}(A \cap B) \leq \min\{\mathbb{P}(A), \mathbb{P}(B)\}$ ($\forall A, B$)). Then such minimum can be further bounded using the basic inequality $(a+b)/2 \geq \sqrt{ab}$ ($\forall a, b \geq 0$). This yields the last term in Theorem 4. We note that at this step, the standard bonus term (1) does not suffice to get an exponential bound. The $\sqrt{T}/n$ design in our new bonus term plays a crucial role.

We next illustrate the proof for $\pi = \text{UCB}$. The proof is in fact simpler. We use the first step in the proof for $\pi = \text{SE}$. For fixed $k$, we also take the same $n_0 = \lfloor x/(2K\Delta_k) \rfloor - 1$. The difference here is that we do not need to define the event $S_k$. When arm $k$ is pulled for the $(n_0+1)$th time, by the design of the UCB policy, there exists some $n$ such that

$$\mu_1 + \frac{\sum_{m=1}^{n} \epsilon_{t_1(m),1} + \sigma\sqrt{\eta T \ln T}}{n} \leq \mu_k + \frac{\sum_{m=1}^{n_0} \epsilon_{t_k(m),k} + \sigma\sqrt{\eta T \ln T}}{n_0}.$$

Now comes the trick. The event is included by a union of two events described as follows:

$$\frac{\sum_{m=1}^{n_0} \epsilon_{t_k(m),k} + \sigma\sqrt{\eta T \ln T}}{n_0} \geq \frac{\Delta_k}{2} \quad \text{and} \quad \frac{\sum_{m=1}^{n} \epsilon_{t_1(m),1} + \sigma\sqrt{\eta T \ln T}}{n} \leq -\frac{\Delta_k}{2}.$$

The probability of each of the two events can be bounded using similar techniques adopted when $\pi = \text{SE}$. In fact, it is implicitly shown in our proof that UCB can yield better constants than SE. We still need to emphasize that when bounding the second event, similar to the argument for $\pi = \text{SE}$, the choice of our new bonus term is crucial.

**Remarks.** For the regret bound, we note that compared to the optimal $\tilde{\Theta}(\sqrt{KT})$ bound, we have an additional $\sqrt{K}$ term. We should point out that the additional $\sqrt{K}$ term is not surprising under the bonus term (4). An intuitive explanation is as follows. Compared to the bonus term (1), we widen the bonus term by a factor of $\sqrt{T/n}$. Among the $K-1$ arms, there must exist an arm such that it is pulled no more than $T/K$ times throughout the whole time horizon. That is, the bonus term of this arm is always inflated by a factor of at least $\sqrt{K}$. The standard regret bound analysis will, as a result, lead to an additional $\sqrt{K}$ factor compared to the optimal regret bound $\tilde{\Theta}(\sqrt{KT})$.

For the tail bound, from our proof road-map, one can see that the tail bound in Theorem 4 is also valid for the pseudo-regret $\sup_\theta \mathbb{P}(R_\theta^\pi(T) \geq x)$. In fact, with some simple manipulations on the inequality, if we let

$$y = \frac{\left(x - 2K - 16\sigma K\sqrt{(\eta \vee 1/\eta)T \ln T}\right)_+}{8\sigma K\sqrt{T}},$$

then for any $x \geq 0$, we can get a neat form (we illustrate this point in the supplementary material)

$$\sup_\theta \mathbb{P}(R_\theta^\pi(T) \geq x) \leq 4K \exp\left(-y^2 \wedge y\sqrt{\eta \ln T}\right).$$

There are two observations:

(a) For any $\eta > 0$, our policy always yields a $O(\sqrt{T})$ expected regret (although with a constant larger than that in the first result in Theorem 4). In fact, notice that for any $x > 0$

$$\mathbb{E}[\hat{R}_\theta^\pi(T)] = \mathbb{E}[R_\theta^\pi(T)] \leq x + \mathbb{P}(R_\theta^\pi(T) \geq x) \cdot T.$$

If we let $x = 2K + C\sigma K\sqrt{(\eta \vee 1/\eta)T \ln T}$ with $C > 0$ being moderately large, then $\mathbb{P}(R_\theta^\pi(T) \geq x) \cdot T = O(1)$. As a result, the worst-case regret becomes

$$O\left(K\sqrt{(\eta \vee 1/\eta)T \ln T}\right).$$

This observation shows that the regret order of our policy design is not related to the parameter $\eta$, as opposed to the standard UCB policy with (1), where a very small $\eta$ may possibly make the UCB policy no longer enjoy a $\tilde{O}(\sqrt{T})$ worst-case regret[1].

---

[1]We believe there is work in the literature that has precisely documented that a very small $\eta$ may possibly make the UCB policy no longer enjoy a $\tilde{O}(\sqrt{T})$ worst-case regret, but we have not been able to identify one. For completeness, we summarize this point as a lemma with a proof in the supplementary material.

(b) If we set $\eta = 1$ and

$$\delta = 4K \exp\left(-(y - \sqrt{\ln T})_+ \sqrt{\ln T}\right) \geq 4K \exp\left(-y^2 \wedge y\sqrt{\ln T}\right),$$

then one can see that for *any* $\delta > 0$, with probability at least $1 - \delta$, the regret of our policy is bounded by

$$O\left(\sigma K\sqrt{T}\left(\sqrt{\ln T} + \frac{\ln(4K/\delta)}{\sqrt{\ln T}}\right)\right) = O\left(\sigma K\sqrt{T}\left(\sqrt{\ln T} + \frac{\ln(1/\delta)}{\sqrt{\ln T}}\right)\right).$$

This partially solve the open question in Section 17.1 of [14] (we've mentioned it in the introduction) up to a logarithmic factor.

## 5 Numerical Experiments

In this section, we provide a brief two-armed numerical experiment result and defer the experiment details and other multi-armed experiment results to the supplementary material. We consider a two armed-bandit problem with $\theta = (0.2, 0.8), \sigma = 1, T = 500$ and Gaussian noise. We test four policies: (i) SE and UCB with the classical bonus design described in (1), and SE_new and UCB_new with the proposed new bonus design in (4). We let $\kappa \triangleq \sigma\sqrt{\eta}$. The tuning parameter $\kappa$ has 4 choices: $\kappa \in \{0.1, 0.2, 0.4, 0.8\}$. We also consider the TS policy assuming the mean reward of each arm $i$ following the prior $\mathcal{N}(0, 1)$ and the sample from each arm $i$ following $\mathcal{N}(\theta_i, \kappa^2)$. That said, we evaluate TS under mis-specified risk parameters. For each policy and $\kappa$, we run 5000 simulation paths and for each path we collect the cumulative reward. We plot the empirical distribution (histogram) for a policy's cumulative reward in Figure 1. Indeed, one can observe that for SE and UCB with (1) and TS, there is a significant part of distribution around 100, indicating a significant risk of incurring a linear regret, especially when $\kappa$ is small. In contrast, with the new design (4), the reward is highly concentrated for every $\kappa > 0$ with almost no tail risk of getting a low total reward. Particularly, when $\kappa = 0.1$, UCB_new achieves both high empirical mean and light-tailed distribution.

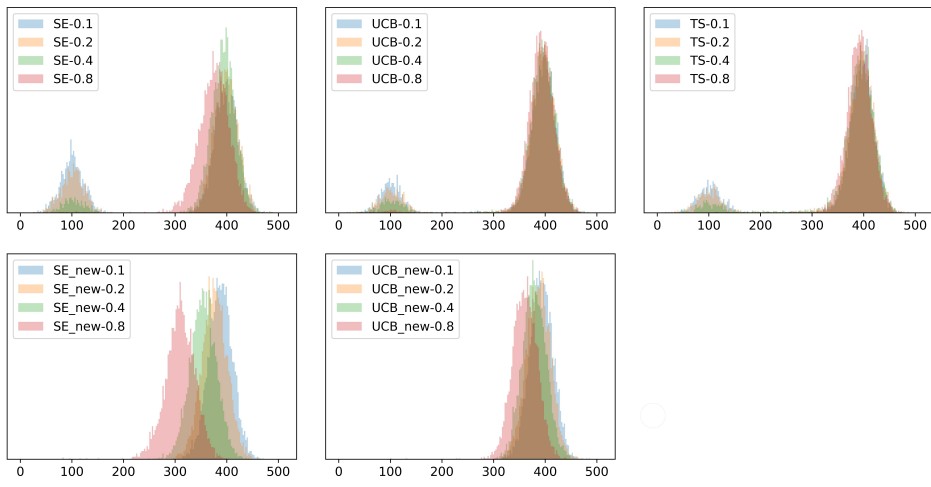

Figure 1: Empirical distribution for the cumulative reward; bottom two are new proposed policies

## 6 Conclusion

In this work, we consider the MAB problem with a joint goal of minimizing the worst-case expected regret and obtaining light-tailed probability bound of the regret distribution. We characterize the interplay among three concepts: light-tailed risk, instance-dependent consistency, and worst-case optimality. We demonstrate that light-tailed risk and instance-dependent consistency are incompatible, and show that light-tailed risk and worst-case optimality can co-exist through a simple new policy design. We discuss insights and generalizations of our results, and study the empirical performance.

An open question is whether we can improve the regret bound in Theorem 4 to $\Theta(\sqrt{KT \ln T})$ and get a probability bound of

$$\ln\left\{\sup_\theta \mathbb{P}(R^\pi_\theta(T) \geq x)\right\} = -\Omega\left(\frac{x}{\sqrt{KT}}\right). \tag{5}$$

Another interesting direction is to generalize our policy design to the setting where noises are not sub-Gaussian, but come from general distributions. In fact, there is a separate line of literature studying bandit problems with non sub-Gaussian noise (see, e.g., [7]). We hope our framework can be integrated with this line of literature to handle certain types of non sub-Gaussian noise, provided with controls on the tail behavior of the random noises.

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
