# OpenReview forum: "A Simple and Optimal Policy Design for Online Learning with Safety against Heavy-tailed Risk"
_NeurIPS.cc/2022/Conference — NeurIPS 2022 Accept_

### Official Review · Reviewer_paa1 · 2022-07-07

**Rating:** 6
**Confidence:** 4
**Soundness:** 2 fair
**Presentation:** 3 good
**Contribution:** 3 good

**Summary:**

The authors consider a novel problem of characterizing the tail behavior of the random regret for algorithms that are designed to solve the multi-armed bandit problem.  In particular, they show that the distribution of regret has a heavy tail in the large T regime for the standard policies, such as Upper Confidence bound (UCB), Successive Elimination (SE), and Thompson Sampling (TS). Subsequently, they propose a simple policy (algorithm) that has light-tailed risk together with minimax optimality for 2 armed bandits. They also extend their analysis to a K-armed setting and support their theoretical claims through experiments. However, all the arguments in this paper only hold for the case when the true reward is observed with additive noise.

**Questions:**

Major:

180-191: The authors can refer to [1] Theorem 2, where the computed instance dependent bound for TS holds for any $\epsilon\in (0,1)$ not a fixed one. So, does this imply that the regret computed using TS policy does not have a heavy tail? Also, it would be useful if the author can also provide an empirical evaluation of TS á la SE and UCB to support their claim in Theorem 1.

401: I think the use of weak law is incorrect here.  How do authors know 'a-priori' that as T increases, the number of samples generated from arm 2 converges to infinity? Without this assumption, the argument doesn't hold.

402. The first inequality is incorrect or has a typo. It should be $\{-N^{\pi}(T)> -(c'-c)T\}$ for the inequality to hold.
In general, many arguments in the proof of Theorem 1 are hand-wavy. For instance, the arguments in 403 hold only for large enough T (see inequality 3 and equality 3, where $\hat \theta$ is considered close to $\theta$). Please make such arguments precise.

Minor:

44-  Please define the term 'instance dependent.' Might not be clear to the readers who are not familiar with the subject.

186, 219-220-  Add citation (with theorems, wherever applicable )

223- Gaussian

[1]: Further Optimal Regret Bounds for Thompson Sampling, Agarwal and Goyal,2013.


**Ethics Review Area:**

["I don’t know"]

**Limitations:**

I authors briefly discuss the limitations in the conclusion section. I think authors can also provide guidance on how to generalize their results to general reward distribution (not just true with additive noise).

**Strengths And Weaknesses:**

1. The authors aptly motivate the problem by citing relevant works and highlighting the novelty of their contribution.
2. The paper is well written with clear definitions.

Weakness:

1. The case of TS in Theorem 1, although it is claimed in the abstract and the introduction; it is not fully investigated.
2. Theorem 2-  Isn't result in (1) here also implies that the policy is consistent using definition 2 in Sec 2.1. If yes, then isn't it a contradiction to the result in Theorem 1?

---

> ### Author Response · Authors · 2022-08-02
> **Response to Reviewer paa1**
>
> We sincerely thank you for your review and very helpful comments.
>
> 1. Response to Weakness 1. The case of TS holds because of (3). More concretely, we have proved that any policy that incurs an instance-dependent $O(\ln T)$ regret must incur a linear regret with probability $\Omega(poly(1/T)) = \exp(-O(\ln T))$. Therefore, standard UCB, SE, and TS all incur heavy-tailed risk, polynomial in $1/T$.
>
> 2. Response to Weakness 2. We hope to clarify that the first result in Theorem 2 does not imply that the policy is consistent. We note that Theorem 2 does not involve consistency. The first result implies a worst-case regret of $\tilde O(\sqrt{T})$, and a worst-case optimal policy may not be consistent. Thus, it is not a contradiction to the result in Theorem 1. We will further clarify the difference between consistency (instance-dependent $O(\ln T)$ expected regret) and worst-case optimality (worst-case $\sqrt{T}$ expected regret).
>
> 3. Response to Major 180-191. We are sorry about the confusion. Theorem 2 in [1] does not contradict our results. First, the bound in [1] holds in an instant-dependent sense, and increases to $+\infty$ when the gap becomes very small, while our definition of light-tailed risk is in the worst-case sense, which means the tail should hold uniformly for all instances. Second, the bound in Theorem 2 of [1] is still an expected regret bound, and the $\epsilon$ is not a probability parameter. As we claimed in Line 190-191, ideally we seek to find a single policy such that it achieves $\tilde O(\sqrt{T\cdot polylog(T/\delta)})$ regret with probability $1-\delta$ for \textit{any} $\delta>0$. We thank the reviewer for suggesting the empirical evaluation of TS and will add that in future versions.
>
> 4. Response to Major 401. Thank you for bringing the issue to us. Yes, the proof requires some additional effort to show $\mathbb P_{\tilde\theta}^\pi(E_T)\to 1$ as $T\to+\infty$. We demonstrate it as follows. Fix any positive integer $N$, we have
> \begin{align*}
> \mathbb P_{\tilde\theta}^\pi(\bar E_T) & = \mathbb P_{\tilde\theta}^\pi(\bar E_T, n_{T, 2} < N) +  \mathbb P_{\tilde\theta}^\pi(\bar E_T, n_{T, 2} \geq N) \\\\
> & \leq  \mathbb P_{\tilde\theta}^\pi(n_{T, 2} < N) + \sum_{n=N}^{+\infty} \mathbb P_{\tilde\theta}^\pi(\bar E_T, n_{T, 2} = n) \\\\
> & \leq \mathbb P_{\tilde\theta}^\pi(n_{T, 2} < N) + \sum_{n=N}^{+\infty}2\exp(-\frac{n\delta^2}{2\sigma^2}).
> \end{align*}
> Thus,
> $$ \liminf_{T}\mathbb P_{\tilde\theta}^\pi(E_T) = 1-\limsup_{T}\mathbb P_{\tilde\theta}^\pi(\bar E_T) \geq 1-\limsup_{T}\mathbb P_{\tilde\theta}^\pi(n_{T, 2} < N)-\sum_{n=N}^{+\infty}2\exp(-\frac{n\delta^2}{2\sigma^2})
> $$
> holds for any $N$. Note that the last term converges to $0$ as $N\to+\infty$. It suffices to show $\mathbb P_{\tilde\theta}^\pi(n_{T, 2} < N)\to 0$ as $T\to+\infty$ for any fixed $N$. Suppose this does not hold, then we can find $p > 0$ and a sequence $\left\\{T(m)\right\\}\_{m=1}^{+\infty}$ such that
> $$
> \mathbb P_{\tilde\theta}^\pi(n_{T(m), 2} < N)>p.
> $$
> Let $M$ be some large number such that $q\triangleq p - N\exp(-\frac{M^2}{2\sigma^2}) > 0$. Then using the change of measure argument, we have
> \begin{align*}
> \mathbb P_{\theta}^\pi(n_{T(m), 2} < N) & = \mathbb E_{\theta}^\pi[1\\{n_{T(m), 2} < N\\}] \\\\
> & = \mathbb E_{\tilde\theta}^\pi\left[\exp\left(n_{T(m), 2}\left(\frac{\tilde\theta_2^2 - \theta_2^2}{2\sigma^2} + \frac{(\theta_2 - \tilde\theta_2)\hat\theta_{T(m), 2}}{\sigma^2}\right)\right)1\\{n_{T(m), 2} < N\\}\right] \\\\
> & \geq \mathbb E_{\tilde\theta}^\pi\left[\exp\left(n_{T(m), 2}\left(\frac{\tilde\theta_2^2 - \theta_2^2}{2\sigma^2} + \frac{(\theta_2 - \tilde\theta_2)\hat\theta_{T(m), 2}}{\sigma^2}\right)\right)1\\{\hat\theta_{T(m), 2} > \tilde\theta_2 - M, n_{T(m), 2} < N\\}\right] \\\\
> & \geq \exp\left(N\left(-\frac{(\tilde\theta_2 - \theta_2)^2}{2\sigma^2} - \frac{M(\theta_2 - \tilde\theta_2)}{\sigma^2}\right)\right)\mathbb P_{\tilde\theta}^\pi(\hat\theta_{T(m), 2} > \tilde\theta_2 - M, n_{T(m), 2} < N).
> \end{align*}
> Note that
> \begin{align*}
> \mathbb P_{\tilde\theta}^\pi(\hat\theta_{T(m), 2} > \tilde\theta_2 - M, n_{T(m), 2} < N) & = p - \sum_{n=1}^{N-1}\mathbb P_{\tilde\theta}^\pi(\hat\theta_{T(m), 2} \leq \tilde\theta_2 - M, n_{T(m), 2} = n) \\\\
> & \geq p - \sum_{n=1}^{N-1}\exp(-\frac{nM^2}{2\sigma^2}) \geq p - N\exp(-\frac{M^2}{2\sigma^2})=q>0.
> \end{align*}
> Therefore, there exists a constant positive probability such that $\pi$ incurs a linear regret under $\theta$, leading to a contradiction.
>
> 5. Response to Major 402. Yes, it should be $-N^\pi(T) \geq -(c'-c)T$, and any $N^\pi(T)$ should be replaced by $-N^\pi(T)$ for Line 402. In Line 399, we have stated that we let $T$ large enough such that $f(T)<(1-2c’)T$. We will make the proof more tidy in our next version.
>
> 6. Response to Minors. We appreciate very much the helpful minor comments. We will update accordingly in the next version.

---

> > ### Comment · Reviewer_paa1 · 2022-08-08
> > **Thank you for your response**
> >
> > I appreciate the authors for addressing all my comments/concerns.
> >
> > In my comment: “However, all the arguments in this paper on.... noise)”. I was, in general, referring to rewards being a sample from a general distribution (not necessarily sub-Gaussian), with/ or without additive structure. However, this is not a big concern since a large class of distributions can be sub-Gaussian. The case of multiplicative error is interesting; I would request authors include a discussion on the same in the appendix.
> >
> > Extending the results in this paper to the non-sub-Gaussian errors is an interesting problem. The authors may consider adding this problem to their paper and cite the work of Bubeck et al. (2013).

---

> > > ### Author Response · Authors · 2022-08-09
> > > **Follow-up Response to Reviewer paa1**
> > >
> > > Thank you for the note and the clarification on the general distribution comment. We have followed the suggestion to add the multiplicative error case to the appendix. For the case with non-sub-Gaussian errors, we have added a brief discussion. We look to add more discussions on non-sub-Gaussian errors and an empirical evaluation for Thompson sampling’s performance in the next version, as suggested by the reviewer. Thanks again for your time and very helpful comments.

---

> ### Author Response · Authors · 2022-08-04
> **Response to Reviewer paa1: The Setup**
>
> We realized that we missed to respond to one constructive comment from the reviewer. We hope to add a response here. The comment was provided as “However, all the arguments in this paper only hold for the case when the true reward is observed with additive noise” and “authors can also provide guidance on how to generalize their results to general reward distribution (not just true with additive noise)”.
>
> We appreciate the reviewer’s constructive comment about additive noise. Based on our understanding, we interpret the comment as follows. “all the arguments in this paper only hold for the case when the true reward is observed with additive noise, but do not hold for the case when the true reward is observed with non-additive noise, say multiplicative noise” and “authors can also provide guidance on how to generalize their results to general reward distribution other the sub-Gaussian distribution”. We hope to respectfully ask for the reviewer’s help on guiding us whether we are on the right track interpreting the comment, and kindly point out our mistakes in our interpretation.
>
> If our understanding was correct, we hope to add the following description. Indeed we have assumed the form of additive noise and did not discuss the multiplicative noise. We hope to briefly review the additive noise notation that we used in the original submission and discuss the connection to multiplicative noise and general reward distribution.
>
> If arm $k$ is pulled in the time period $t$, we have written in the submission that the observed reward is $r_{t,k} = \theta_k + \epsilon_{t,k}$, where $\theta_k$ is the true expected reward and $\epsilon_{t,k}$ represents a mean zero random noise. We have assumed this mean-zero random noise to be $\sigma$ sub-Gaussian, which appears to be a standard assumption in the literature and includes distributions such as Gaussian and Bernoulli.
>
> Instead of modeling the random reward using additive noise, one may naturally alternatively model the random reward using multiplicative noise. That is, if arm $k$ is pulled in the time period $t$, the observed reward $r_{t,k} = \theta_k \cdot  \epsilon’_{t,k}$ where $\theta_k$ is the true expected reward and $\epsilon’_{t,k}$ represents a mean-one random noise. Some assumptions are needed to describe the distribution of the multiplicative noise $\epsilon’_{t,k}$. If the multiplicative noise $\epsilon’_{t,k}$ is $\sigma$ sub-Gaussian, then if we rewrite the observed reward as $r_{t,k} = \theta_k \cdot  \epsilon’_{t,k} = \theta_k + \theta_k( \epsilon’_{t,k} - 1)$, the multiplicative noise can be transformed into an additive noise with the additive part given by $\theta_k( \epsilon’_{t,k} - 1)$. If $\epsilon’_{t,k}$ is $\sigma$ sub-Gaussian, then the random term $\theta_k( \epsilon’_{t,k} - 1)$ is sub-Gaussian, and therefore the multiplicative case may be transformed into an additive case with moderate modifications. If otherwise,  $\epsilon’_{t,k}$ is not $\sigma$ sub-Gaussian, then it means that the random term $\theta_k( \epsilon’_{t,k} - 1)$ is likely also not sub-Gaussian. This would project the question back into what would happen to multi-armed bandit theory and algorithm design if the additive noise is not sub-Gaussian. That is, the noise comes from general distributions. In fact, there is a separate line of literature studying bandit problems with non sub-Gaussian noise (see, e.g.,  Bubeck et. al. (2013)). We think our framework can be integrated with this line of literature to handle certain types of non sub-Gaussian noise, provided with controls on the tail behavior of the random noises.
>
> We hope to respectfully ask for the reviewer’s help on guiding us whether we are on the right track interpreting the comment, and kindly point out our mistakes in interpretation. We like to express our sincere appreciation for this clarification.
>
>
> Bubeck, Sébastien, Nicolo Cesa-Bianchi, and Gábor Lugosi. "Bandits with Heavy Tail." IEEE Transactions on Information Theory 59.11 (2013): 7711-7717. https://arxiv.org/abs/1209.1727

---

### Official Review · Reviewer_tcDL · 2022-07-10

**Rating:** 7
**Confidence:** 4
**Soundness:** 4 excellent
**Presentation:** 4 excellent
**Contribution:** 4 excellent

**Summary:**

The paper studies tail behaviors of bandit problems. On the negative side, it proves that any algorithm that achieves instance-dependent consistency (logarithmic regret) must exhibit heavy-tailed regret. On the positive side, it shows that it is possible to achieve light-tailed and worst-case optimal regret (on the order of $\sqrt{T}$) at the same time. This is demonstrated by Successive Elimination and Upper Confidence Bound equipped with a novel bonus. The paper also conducts numerical experiments to verify its theoretical claims.

**Questions:**

N/A

**Strengths And Weaknesses:**


Strengths
- The theoretical results of the paper appear solid and sound. In particular, both the negative and positive results give a fairly complete characterization of tail behavior of the studied bandit algorithms. The relative novelty of the paper compared to existing literature is clearly discussed and looks significant.
- The numerical experiments help support and visualize the theoretical contributions of the paper.

Weaknesses
- There is potential for improvement in some of the bounds (also discussed in Section 6), but I think that the current results are significant enough for it to be publishable.

---

> ### Author Response · Authors · 2022-08-02
> **Reponse to Reviewer tcDL**
>
> We sincerely thank you for your review and very helpful comments.
>
> Response to weakness: We appreciate the reviewer’s comment. As is suggested, there is indeed potential improvement on the tail bound, especially related to the dependence on $K$. To improve this dependence on $K$, a possible direction is to shrink the bonus by a factor of $\sqrt{K}$. However, it may require significant additional efforts because the shrinkage will make the current analysis fail, especially when we consider the situation where $S_k$ happens (the bonus becomes too small, see., e.g., Line 259-260, 490, 496).

---

### Official Review · Reviewer_3Vuu · 2022-07-11

**Rating:** 6
**Confidence:** 4
**Soundness:** 3 good
**Presentation:** 3 good
**Contribution:** 3 good

**Summary:**

This work proves that an algorithm cannot incur an optimal expected regret and achieves light-tailed meanwhile. It shows the trade-off between these two targets in Theorem 1, and studies the performances of the new SE and UCB algorithms theoretically and numerically.

**Questions:**

1. In Theorem 1, it states that "If a policy is instance-dependent consistent, then it can never be light-tailed. Moreover, if $\pi = $ SE or UCB with ... ...". I can see this statement is supported by the proof of Theorem 1, but the exact equation for the first sentence should be presented here. I am interested at what equation this conclusion is drawn from.
2. Only the bounds for SE and UCB algorithms are extended in the $K$-armed bandit setting, what about the discussion that "If a policy is instance-dependent consistent, then it can never be light-tailed"? What is the difficulty to prove the same result in the $K$-armed bandit setting?
3. Minor mistake: Line 130 --- "one only need to consider ... ..." should be "one only needs to consider ... ..."

Overall, I believe this observation makes some contribution to the bandit area. However, I would expect some clarification about the difficulty to analyze the $K$-armed bandit setting.

**Limitations:**

I don't see any societal issue in this work.

**Strengths And Weaknesses:**

Strengths: The work is in general easy to follow. The findings for $2$-armed bandits are quite complete and some of them are extended to the $K$-armed bandits.

Weaknesses: More theoretical results have been presented for the $2$-armed bandits than the $K$-armed bandits. The authors may consider to explain the difficulties to derive all "similar" results for the K-armed bandits. More clarifications are in the "Question" section.

---

> ### Author Response · Authors · 2022-08-02
> **Reponse to Reviewer 3Vuu**
>
> We sincerely thank you for your review and very helpful comments.
>
> 0. Related to the reviewer’s summary, we hope to clarify that our submission proves that an algorithm cannot incur an instance-dependent optimal expected regret at the order $\ln T$ and achieves light-tailed risk meanwhile. Additionally, our results (Theorem 2 and 4) prove that an algorithm can achieve a worst-case optimal expected regret at the order $O(\sqrt{T})$ and achieves light-tailed risk meanwhile.
>
> 1. Response to Question 1: The conclusion is drawn from the equations on Line 405-409. We follow our definition of light-tailed risk, and show that for any $\beta>0$, the log probability of incurring a linear regret cannot decay slower than any polynomial $T^\beta$.
>
> 2. Response to Question 2: Since $2$-armed bandit is a special case of the general $K$-armed bandit problem, a negative result held under the $2$-armed bandit setting implies that it is held under the general setting. As a concrete example, one can embed the two-armed bandit setting into a $K$-armed bandit one by creating $K-2$ dummy arms with reward $0$ and noise $0$. Then the policy should always pull the first two arms and ignore the remaining ones, which is essentially equivalent to the $2$-armed bandit setting.
>
> 3. Response to Question 3: We thank the reviewer for pointing it out. We will correct the sentence in the future version.

---

> > ### Comment · Reviewer_3Vuu · 2022-08-08
> > **Rebuttal is acknowledged**
> >
> > Thanks the authors for clarification. I increased the rating.

---

> > > ### Author Response · Authors · 2022-08-09
> > > **Follow-up Reponse to Reviewer 3Vuu**
> > >
> > > Thank you for the note. Thanks again for your time and very helpful comments.

---

### Official Review · Reviewer_tyUi · 2022-07-13

**Rating:** 7
**Confidence:** 4
**Soundness:** 4 excellent
**Presentation:** 4 excellent
**Contribution:** 3 good

**Summary:**

This paper studies the tail properties of the regret attained by multi-armed bandit schema. The first result shows that any consistent policy must suffer a heavy tailed regret distribution in the sense that the chance of incurring linear regret decays at best polynomial with the horizon $T$, thus generalising a recent result of Fan and Glynn.

The authors complement this negative result by developing simple schema that, for bandits with subGaussian noise, a) attain nearly minimax optimal expected regret bound rates of $O(K\sqrt{T \log T}),$ and b) have light tailed regret distributions in the sense that simultaneously for all $\delta, $ the chance that the regret exceeds some $\Omega(K  \sqrt{T \log T} ( 1 + \log(1/\delta)/\log T))$ quantity is bounded by $\delta$. This represents significant progress on the open question of finding minimax bandit schema with good high probability regret behaviour. The schemes achieving this behaviour are quite simple - successive elimination or UCB, but run with a width of the form $\sqrt{T}/n$ instead of the usual $\sqrt{1/n}$, the results follow from a careful but elementary analysis.

Additionally, a small simulation study demonstrates both the heavy tails of the standard methods, and the light tails of the proposed schema.

**Questions:**

This is more a suggestion than a question: given the results, it is natural to expect a trade-off between how small one can drive instance dependent regrets and the tail properties. Do you have a sense of what this trade-off is?

Concretely, slightly modify the definitions in section 2.1 to say that a method is $\rho$-light tailed if the property of definition 1. holds for all $\beta \le \rho,$ and $\alpha$-consistent if the property of definition 2. holds for all $\beta > \alpha$. Clearly every scheme is $0$-light tailed, so we can define $\rho(\alpha)$ as the largest $\rho$ such that there is a $\alpha$-consistent, $\rho$-light tailed method. Then Thm. 1 can be interpreted as $\rho(0) = 0,$ and Thm. 3 & 4 as saying $\rho(1/2) = 1/2$. I think characterising the behaviour for intermediate $\alpha$ would be an interesting question (and the techniques of Thms 3,4. should at least yield bounds on $\rho(\alpha)$).

**Limitations:**

Barring the discussion of $K$ v/s $\sqrt{K}$ mentioned previously, this is fine.

**Strengths And Weaknesses:**

In my opinion, the paper studies an interesting problem in a simple and elegant way. The poor tail properties of asymptotically optimal bandit schema have long been folklore, and the negative result fleshes out this property in considerable generality. The schemes proposed and their analyses are elegant and simple. I find the width schedule of $\sqrt{T}/n$ quite surprising, in terms of both the very aggressive initial exploration and the aggressive step-off to exploitation it represents. The exposition is clear and precise, and the results are well motivated, interesting, and well explained.

I think the only real quibble I have with the paper is that a clear discussion of the $K$ v/s $\sqrt{K}$ distinction is not present - at most this is weakly acknowledged when defining minimax optimality. I think the paper would benefit from a clear acknowledgement of this gap, a frank description of where the analysis breaks down when shooting for the stronger $\sqrt{K}$-dependence, and a discussion on the authors' opinions on how close to this one can get by refining the techniques of this paper.

Overall I think this is a solid paper, and should be of meaningful interest to the online learning community.

---

> ### Author Response · Authors · 2022-08-02
> **Reponse to Reviewer tyUi**
>
> We sincerely thank you for your review and very helpful comments.
>
> 1. Regarding the trade-off between instance dependency and light tail, your suggestion is quite valuable to us. Under your suggested definition, our guess is that $\rho(\alpha)=\alpha$ always holds for $\alpha\in[0, 1/2]$. That is, the more inconsistent a policy is, the more light-tailed risk a policy can be. Establishing this concrete relationship requires some effort and we leave it for future work. We think the techniques in Thms 2, 3 can lead to tight bounds.
>
> 2. Regarding the suboptimal dependence of $K$ in our bound, we add some discussions as follows. First, the current policy design can only yield a $\tilde O(K\sqrt{T})$ regret. An intuitive explanation is as follows. Compared to the bonus term (1), we widen the bonus term by a factor of $\sqrt{T/n}$. Among the $K-1$ arms, there must exist an arm such that it is pulled no more than $T/K$ times throughout the whole time horizon. That is, the bonus term of this arm is always inflated by a factor of at least $\sqrt{K}$. The standard regret bound analysis will, as a result, lead to an additional $\sqrt{K}$ factor compared to the optimal regret bound $\tilde\Theta(\sqrt{KT})$. Second, a possible direction is to shrink the bonus by a factor of $\sqrt{K}$. However, this shrinkage will make the current analysis fail, especially when we consider the situation where $S_k$ happens (the bonus becomes too small, see., e.g., Line 259-260, 490, 496). Additional efforts on the analysis would be needed.

---

### Author Response · Authors · 2022-08-02
**Literature Update**

We hope to write to the review panel about some imprecise descriptions related to a literature work [8].

In our submitted main paper, we wrote that “The previous works do not have this positive result.”

After the submission of our work, we realized that a more recent version of [8] (which can be found on https://arxiv.org/pdf/2109.13595.pdf) included a positive result. In [8], the authors provide modified UCB algorithms that can ensure a desired polynomial rate of tail risk, which in the meanwhile makes the algorithms more robust to mis-specifications. We will acknowledge this contribution to the literature.

In our submitted paper, we wrote that “We show that a wide range of policies, including the standard UCB and SE policy, and the TS policy, suffer from heavy-tailed risk.” and that “[8] show that for an information-theoretic optimized bandit policy, the probability of incurring a linear regret is at least $\Omega(1/T)$. We generalize their result by stating that a much larger family of policies with consistency is heavy-tailed.”

Based on the more recent version of [8], we will update our statements as follows. “Recently, [8] showed that information-theoretic optimized bandit policies as well as general UCB policies suffer from some serious heavy-tailed risk; that is, the probability of incurring a linear regret slowly decays at a polynomial rate of $1/T$, as $T$ (the time horizon) increases. Inspired by the results in [8], we adapt their change of measure argument to show that any policy that incurs an instance-dependent $O(\ln T)$ regret must incur a linear regret with probability $\Omega(poly(1/T)) = \exp(-O(\ln T))$.”


[8] Fan L, Glynn PW (2021b) The fragility of optimized bandit algorithms. arXiv preprint arXiv:2109.13595

---

### Meta-Review · Area_Chair_G1hA · 2022-08-24

**Recommendation:** Accept
**Confidence:** Certain

**Metareview:**

The reviewers came to consensus that this paper makes a good contribution to the study on the tail behavior of the regret of bandit problems. I agree with these opinions and please polish the paper so that the minor concerns raised by the reviewers become clear in the final version.

**Award:**

No

---

### Decision · Program_Chairs · 2022-09-14

Accept